# College Students' Entrepreneurial Mindset: Educational Experiences Override Gender and Major

**Eunju Jung [1],\* and Yongjin Lee [2],\***

[1]  Graduate School of Education, Sejong University, Seoul 05006, Korea
[2]  Department of Liberal Arts, Hansei University, Gunpo, Gyeonggi-do 15852, Korea
\*  Correspondence: doduli@sejong.ac.kr (E.J.); eduist@hansei.ac.kr (Y.L.)

**Abstract:** Entrepreneurship education has been popularly adopted in higher education contexts. Although evidence-based implementations of such education are widely acknowledged as beneficial, valid assessments of it are sparse. One possible outcome of entrepreneurship education is a change in students' entrepreneurial mindset, which can be measured by the recently validated College Students' Entrepreneurial Mindset Scale (CS-EMS). However, this scale awaits evidence regarding measurement invariance. This study aims to (1) examine measurement invariance of the CS-EMS; (2) compare the latent and observed means across groups based on gender, major, and educational experiences; and (3) investigate the conditional effects of the three grouping variables. Using data from 317 Korean college students' survey responses, we conducted sequential tests of factorial invariance and latent mean comparisons using multiple-group confirmatory factor analysis. Additionally, the conditional effects of the gender, major, and educational experiences were tested by structural equation modeling. The results indicate that strict invariance held for the groups compared by either gender or educational experiences, while scalar invariance held between the engineering and non-engineering groups. While the male, engineering, and educational experience groups generally scored higher on both the latent and observed sub-scales, the results of the conditional effects of grouping variables indicated that educational experiences mattered most. One practical implication for the educators is that the CS-EMS is a promising assessment tool for addressing the effectiveness of entrepreneurship education, especially when the targeted educational goals are any of its sub-constructs.

**Keywords:** entrepreneurial mindset; college students; gender; engineering; educational experience; measurement invariance; latent mean comparisons

## 1. Introduction

Since the Harvard Business School's pioneering entrepreneurship class was offered in 1947, entrepreneurial education has been expanded to diverse disciplines in higher education [1–4]. In addition, entrepreneurship education has gained global popularity among both undergraduate and graduate students [1,5]. It is also highly valued in Korea, and such courses are not uncommon in higher education curricula in diverse disciplines [6–9]. The wide dissemination of entrepreneurship education can be attributed to its expected beneficial outcomes, such as improved skills, knowledge, and attitudes related to venture creation [10], increased self-employment and ability to launch start-ups [10,11], and eventually economic growth [12]. Yet, the expected benefits are not limited to the realm of business, management, and the economy, especially in the context of higher education. The scope of entrepreneurship education has been extended to embrace broader educational goals for college students, such as improved career self-efficacy, career adaptability, project-management skills, self-regulation, and intrapreneurship in certain professional fields after graduation [13]. Due to the increasing volatility and uncertainty in job market and various career fields, college students today face

more challenges than their counterparts in the past [14]. They are more likely to encounter a shortage of stable life-long careers, more project-based short-term jobs, and jobs replaced by artificial intelligence (AI). As a result, they might need more career adaptability to allow them to pursue multiple different career paths. For them, an entrepreneurial mindset, which might enhance their career adaptability, would be a valuable asset in today's era of uncertainty and fluctuation in the workplace [14].

Participating in the broadening of entrepreneurship education, Korean universities have provided diverse educational programs ranging from short-term, intensive, experiential, and extracurricular programs [13], to formal classes lasting one semester [15]. The educational goals range from the promotion of creativity to teamwork, communication skills, product development, and opportunity identification [13]. In addition, an enhanced entrepreneurial mindset was expected in most of the programs. However, the effectiveness of entrepreneurship education has not been thoroughly studied; to date, educational effectiveness has been measured by only one or a couple of entrepreneurial intention questions in many studies (e.g., [16–19]).

This can be mainly attributed to the lack of quality-assured assessment tools to measure various aspects of educational outcomes in higher education settings. Among the available measurement instruments, the Builder Profile [20] and the Global University Entrepreneurial Spirit Students' Survey (GUESSS [21]) had little evidence of reliability and validity. Although the Individual Entrepreneurial Orientation (IEO) [22,23] and the Entrepreneurial Mindset Profile (EMP) [24] thoroughly examined reliability of and evidence for multiple validity issues (e.g., construct validity, criterion-related validity, predictive validity, etc.), their measurement invariance has never been investigated.

The College Students' Entrepreneurial Mindset Scale (CS-EMS) [25], a recently developed and validated assessment, is promising for systematic measurement of the sub-constructs of *innovativeness*, *need for achievement*, *risk-taking*, *autonomy*, and *proactiveness*, which are the mindsets that are targeted for improvement across a wide spectrum of entrepreneurship classes. Yet, the measurement invariance of the CS-EMS across gender, major, and educational experiences has never been examined, and it is unknown which grouping variable has the most influence on the sub-scales of the CS-EMS.

To fill the void in the literature on entrepreneurship in higher education, this study was designed to pursue the following three goals. First, we tested four increasingly stringent measurement invariance models (i.e., configural, metric, scalar, and strict invariance models) of the CS-EMS across gender, major, and experience groups using the multi-group confirmatory factor analysis (MG-CFA) framework. Second, we examined the latent and observed mean differences in the sub-scales of the CS-EMS across the studied groups only if scalar invariance had been established. Third, we investigated the conditional effects of the three grouping variables (i.e., *gender*, *major*, and *experience*) using the structural equation modeling framework.

We expect that the findings of the current study will be able to guide educators when they use assessment tools to compare groups. Specifically, entrepreneurship educators will learn that cross-group comparisons based on observed or latent means should be preceded by a measurement invariance test [26–29]. In addition, the findings from the cross-group mean comparisons reveal the compared groups' current status regarding the entrepreneurial mindset, and educators might be able to design their entrepreneurship education programs with more emphasis on the areas that need improvement in particular gender [30–32], major [30,31], or experience groups [33,34]. Moreover, the findings based on the conditional effects of the grouping variables imply the necessity of entrepreneurship education for college students if educational experiences with entrepreneurship are found to be the factor with the most influence on the CS-EMS sub-scales. Last, but not least, we expect that the CS-EMS will serve as an important assessment tool for reliably and validly measuring the effects of entrepreneurship education in cases where the targeted educational objectives are related to any of the sub-constructs of the CS-EMS [35,36].

In the remainder of this manuscript, we first review the previous studies that are most relevant to the current study in terms of four themes: concepts of entrepreneurship and entrepreneurial mindset, currently available assessment tools and their limitations, measurement invariance, entrepreneurship

education for college students, and issues related to gender or major differences. Next, we describe the characteristics of the participants, the CS-EMS instrument, and the analytic procedure, providing information on the materials and methods. Then, we illustrate the results of the current study for the measurement invariance test, cross-group mean comparisons, and conditional effects of the gender, major, and experience variables. Subsequently, we discuss the findings, implications, limitations, and suggestions for future studies, followed by the conclusions of the study.

## 2. Literature Review

### 2.1. Concepts of Entrepreneurship and Entrepreneurial Mindset

Researchers have defined entrepreneurship as a compound construct with various assets. Venkataraman [37,38] asserted that entrepreneurship refers to an activity that involves the discovery, evaluation, and exploitation of opportunities to introduce new goods and services, ways of organizing, processes, and raw materials [38]. Based on Miller and Friesen's work [39], the concepts of innovativeness, risk-taking, and proactiveness are commonly used to characterize and test entrepreneurship [40,41]. In addition to those three elements, Lumpkin and Dess [42] identified two more dimensions, autonomy and competitive aggressiveness, that are used to conceptualize entrepreneurial orientation. Entrepreneurial orientation has emerged as a key construct in the entrepreneurship literature. It has been viewed as a characteristic of organizations that can be measured by looking at the top management's entrepreneurial style, as evidenced by the firms' strategic decisions and operating management philosophy [43]. This concept of entrepreneurship focuses more on entrepreneurial behaviors, including seeking, identifying, grasping or creating opportunities, taking the initiative, solving problems, organizing and coordinating resources, networking effectively, combining things innovatively, taking calculated risks, and acting proactively in complex situations [44–46].

Entrepreneurship has been also defined as a mental attitude deeper than an intent to merely create a business. It requires application of energy and passion to create and implement new ideas and creative solutions [5]. Bosman and Fernhaber [47] describe the entrepreneurial mindset as an inclination toward entrepreneurial activities. A mindset is an individual's mental attitude or state that predetermines one's responses to and interpretations of a given situation [31]. An entrepreneurial mindset includes an individual's willingness to blend risk-taking, creativity, and innovation with the intention of creating value as well as an individual's ability to plan and manage projects in order to achieve objectives [47–49]. It relates to being dynamic, flexible, and self-regulating in an uncertain environment [44,45]. The entrepreneurial mindset develops over time and requires practice [47]. This supports individuals during daily life and makes employees more aware of the context of their work and better able to seize opportunities [47]. Thus, entrepreneurial-minded learning has received increased interest as a pedagogical approach within the higher education field [30,31].

When discussing entrepreneurship, the literature separates entrepreneurial mindsets from entrepreneurial behaviors [50]. Entrepreneurial mindsets refer to the abilities and general attitude of an individual, while entrepreneurial behaviors are made evident through the individual's actions. Both entrepreneurial mindsets and behaviors are valid concepts not only when dealing with business but also in all human activities [50]. Because entrepreneurship is not only about knowing facts but also a way of thinking and acting [46], recently, higher education programs have defined entrepreneurship broadly and included enterprising behaviors outside the business context [46,51–53].

### 2.2. Assessments for Entrepreneurial Mindsets

The literature has described several assessment instruments that are designed to measure an individual's entrepreneurial orientation and mindset. However, previous measures for entrepreneurial characteristics lack quality evidence, justifying the need for a validated measure of the entrepreneurial mindset. Some instances of instruments are reviewed as follows.

First, Badal and Struer [20] developed Builder Profile 10 to identify individual characteristics that are associated with building a successful business. The instruments include 30 items representing ten characteristics (determination, independence, confidence, delegator, risk, profitability, relationship, disruptor, knowledge, and selling). Evidence regarding its construct validity has never been examined, although its validity has been extensively investigated in relation to other variables. In addition, to our knowledge, it has never been validated for college students and has only been validated with high school and entrepreneur samples in the US.

Second, the Global University Entrepreneurial Spirit Students' Survey was developed in 2006 and designed to measure university students' perceptions of entrepreneurs (11 items) and their entrepreneurial competencies (seven items) in addition to entrepreneurial intentions. Although it has been widely used internationally until recently [21,54], its reliability and validity have never been tested.

Third, the Individual Entrepreneurial Orientation (IEO) scale, which has ten items, was developed by Bolton and his colleagues [22], and they found that the three correlated-factor structure was tenable based on validation with 1,100 university students. The three sub-factors were innovativeness, risk-taking, and proactiveness. Popov and colleagues [23] recently examined the construct validity of the IEO scale with Serbian college students and adults, and their results also supported the three correlated-factor structure of the ten items. However, neither study considered the measurement invariance of the IEO.

Fourth, the Entrepreneurial Mindset Profile (EMP) [24] was developed in 2015, and it was constituted of 14 dimensions with 72 items. Among the 14 dimensions, seven dimensions (i.e., independence, limited structure, non-conformity, risk acceptance, action orientation, passion, and need to achieve) represented traits of entrepreneurs, while the remaining seven dimensions (i.e., future focus, idea generation, execution, self-confidence, optimism, persistence, and interpersonal sensitivity) represented skills for entrepreneurs. They provided validity evidence based on the internal structure of the items and their relations to other variables. Although they compared the sub-scale scores of the EMP across gender, they did not consider measurement invariance before making a cross-group mean comparison.

*2.3. Measurement Invariance*

Measure invariance is an important issue, especially when a researcher wants to make cross-group comparisons using a measurement instrument consisting of multiple items that are assumed to have a smaller number of factors underlying them [27,28,55,56]. The core question in measurement invariance is whether the assessment or measurement in use operates in the same way across different groups based on either demographic characteristics (e.g., gender [57,58], nationality [57,59], language in use [55], etc.) or certain artifactual categorizations (e.g. experimental vs. treatment group [60,61]; pre- vs. post-measurement [62,63]; internet-based test vs. paper-and-pencil test [64]).

One of the most widely used methods to test measurement invariance is a multiple-group confirmatory factor analysis (MG-CFA) model which is a multi-group extension of a confirmatory factor analysis model [26–29,56,65]. Measurement invariance tested under the MG-CFA framework is also called factorial invariance, and it is well-known for its flexibility in examining every measurement parameter: factor loading ($\lambda$), intercept ($\tau$), and unique variance ($\theta$) [26,29,56]. The conventional way to test measurement invariance involves four sequential steps to evaluate increasingly constrained models – from configural invariance to strict invariance – across the studied groups [26–29]. Configural invariance indicates that the same factor structure holds between the groups while all measurement parameters are freely estimated for each group, which implies that the groups interpret a given set of items using equal conceptual grounding [37,55,66]. Once configural invariance is established, metric invariance is tested by imposing equality constraints on all factor loadings between the groups. Under the condition of metric invariance, the strength of the relationship between a factor and items belonging to the factor is equivalent across the groups [28,55,66]. Upon the established metric

invariance, strict invariance is tested by equally constraining all sets of intercepts between groups. Scalar invariance can be interpreted as indicating that the origin of the item score is the same across the groups [27,28,55,67]. Finally, strict invariance is tested by adding equality constraints on the pair of unique variances between the groups upon the established scalar invariance model [68]. The status of strict invariance can be interpreted as indicating that the degree of errors is equivalent across groups [29]. Among the four measurement invariance conditions, the scalar invariance condition is necessary to compare the latent and observed means across groups [26,27,29], and thus, we drew the following hypotheses:

**Hypothesis 1 (H1).** *The CS-EMS presents at least scalar invariance across gender, major, and experience groups.*

**Hypothesis 1a (H1a).** *The CS-EMS presents at least scalar invariance between the male and female groups.*

**Hypothesis 1b (H1b).** *The CS-EMS presents at least scalar invariance between the engineering and non-engineering groups.*

**Hypothesis 1c (H1c).** *The CS-EMS presents at least scalar invariance between the experience and no-experience groups.*

*2.4. Entrepreneurship Education for College Students*

Entrepreneurship education mainly focuses on the development of certain beliefs, values, and attitudes, with the aim of causing individuals to consider entrepreneurship as an attractive and valid alternative to paid employment or unemployment [34,69]. Since the early 2000s, entrepreneurship education programs in higher education have grown rapidly and globally [1,2,5,70] in an effort to promote entrepreneurial outcomes [36]. The global interest in entrepreneurship education is a result of the association between entrepreneurship and economic growth, which has motivated policymakers to focus on cultivating and sustaining entrepreneurship [71]. Entrepreneurship education is a major approach to developing entrepreneurial intentions, mindsets, and behaviors [72]. However, the research on the impact of entrepreneurship education on entrepreneurial mindsets or intentions has yielded mixed results [1,35]. The literature has suggested that it is important to analyze the impact of entrepreneurship in gender-specific and pedagogy-specific manners [1]. In the following subsection, the studies on gender differences, major differences, and differences based on educational experiences in entrepreneurship are introduced.

2.4.1. Comparisons Based on Gender

Past research on gender differences in entrepreneurship has typically found that females are more conservative in entrepreneurial activities than males [73,74]. The image of the entrepreneur has traditionally been masculinized and rooted in masculine discourse [75]. Moreover, research has found that for women who work in gender incongruent occupations dominated by men, the experience of discrimination has a negative association with their well-being [76].

Research on the impact of entrepreneurship education on students' intention and mindset has reported gender-specific differences [77]. With students who have less exposure to entrepreneurship, the general effect of entrepreneurship education tends to be positive because participation in the programs usually increases their entrepreneurial intentions, attitudes, and self-efficacy [78]. Nowiński et al. [79] investigated whether entrepreneurial education contributes to the entrepreneurial intentions of university students in the Czech Republic, Hungary, Poland, and Slovakia. They indicated that although women generally have lower entrepreneurial intentions and display lower levels of entrepreneurial self-efficacy, they benefit from entrepreneurship education more than men do [79]. However, emerging literature shows that the relations between gender and the entrepreneurial mindset are more complex and multi-faceted. For example, Majumdar and Varadarajan [80] investigated the

entrepreneurial mindset of women in the Arab world and suggested that the propensity for future entrepreneurship does not depend on gender; rather, it depends on factors like creativity, motivation, and awareness. An educational system that lacks a supportive environment and concrete initiatives can deeply affect female students, causing them to fear engaging in entrepreneurship [81]. Although efforts to promote an entrepreneurial mindset within society have increased, there has still been little attention on assessment and analysis of the entrepreneurial mindset amongst female students in the context of higher education. In addition, the results from the previous studies generally indicate that the females showed a lower level of entrepreneurial attitudes, intentions, and behaviors, thus we suggest the following hypotheses:

**Hypothesis 2 (H2).** *The male group scores higher on each of the five sub-constructs of the CS-EMS than the female group.*

**Hypothesis 2a (H2a).** *The male group scores higher on innovativeness than the female group.*

**Hypothesis 2b (H2b).** *The male group scores higher on need for achievement than the female group.*

**Hypothesis 2c (H2c).** *The male group scores higher on risk-taking than the female group.*

**Hypothesis 2d (H2d).** *The male group scores higher on autonomy than the female group.*

**Hypothesis 2e (H2e).** *The male group scores higher on proactiveness than the female group.*

2.4.2. Comparison Based on Major: Engineering vs. Non-Engineering

Specifically, engineering education institutes play an important role in entrepreneurial development [14]. Engineers often take positions in which entrepreneurship is highly valued because they work in areas in which technological development is moving very quickly. As entrepreneurship serves as an integral part of the economy, engineers need to develop an entrepreneurial mindset through authentic educational experiences [82]. Thus, engineering education institutes have been interested in developing an academic entrepreneurship education community through the development of engineering-specific entrepreneurship centers and programs [83].

In South Korea, there is strong pressure to develop entrepreneurship and innovation competencies in engineering education [14]. The industry has influenced the process to improve this part of engineering education, which in turn has prompted the government to consider entrepreneurship education to be crucial [14]. In the accreditation process for engineering education, universities should prove that their curricula, including capstone design courses, promote students' entrepreneurial mindset, and skills. Capstone design courses often guide students from the problem identification stage through prototyping, with a heavy focus on technological feasibility and an entrepreneurial mind. While the creation of engineering entrepreneurship programs seems to address the need for reforms in undergraduate engineering programs, such programs usually measure output metrics, such as enrollment and degrees, as opposed to evidence of the program's impact on each individual student's mindset [83]. To our knowledge, no study has directly compared the difference in entrepreneurial mindset among different majors. However, considering the efforts to promote students' entrepreneurial attitudes, intentions, and behaviors made by engineering disciplines we suggest the following hypotheses regarding major difference:

**Hypothesis 3 (H3).** *The engineering group scores higher on each of the five sub-constructs of the CS-EMS than the non-engineering group.*

**Hypothesis 3a (H3a).** *The engineering group scores higher on innovativeness than the non-engineering group.*

**Hypothesis 3b (H3b).** *The engineering group scores higher on need for achievement than the non-engineering group.*

**Hypothesis 3c (H3c).** *The engineering group scores higher on risk-taking than the non-engineering group.*

**Hypothesis 3d (H3d).** *The engineering group scores higher on autonomy than the non-engineering group.*

**Hypothesis 3e (H3e).** *The engineering group scores higher on proactiveness than the non-engineering group.*

2.4.3. Comparison Based on Educational Experiences in Entrepreneurship

Regarding the impact of entrepreneurship education, Bae and colleagues' meta-analytic review [36] found a significant correlation between entrepreneurship education and entrepreneurial intentions. They emphasized that it is important to consider the significant impact of moderators, such as the attributes of entrepreneurship education, differences between students, and cultural values, on entrepreneurial intentions. Most studies suggest a positive link between the educational program and students' entrepreneurial intentions, attitude, knowledge, and skills [84–87], but some articles report results that are not significant or negative. For example, Lanero, et al. [88] reported that there is no significant link between entrepreneurship education and entrepreneurial attitudes among Spanish students. Also, Mentoor and Friedrich [89] found a negative link between educational experiences and attitudes toward entrepreneurship among South African students. Indeed, there is still limited attention given to the impact of entrepreneurship education and the quality-assured assessment tools to measure various aspects of educational outcomes within the context of cross-cultural and academic majors [2]. Therefore, we aim to confirm the influence of educational experience in entrepreneurship with the validated assessment tool, and suggest the following hypotheses:

**Hypothesis 4 (H4).** *The group with educational experiences in entrepreneurship scores higher on each of the five sub-constructs of the CS-EMS than the group without such experiences.*

**Hypothesis 4a (H4a).** *The group with educational experiences in entrepreneurship scores higher on innovativeness than the group without such experiences.*

**Hypothesis 4b (H4b).** *The group with educational experiences in entrepreneurship scores higher on need for achievement than the group without such experiences.*

**Hypothesis 4c (H4c).** *The group with educational experiences in entrepreneurship scores higher on risk-taking than the group without such experiences.*

**Hypothesis 4d (H4d).** *The group with educational experiences in entrepreneurship scores higher on autonomy than the group without such experiences.*

**Hypothesis 4e (H4e).** *The group with educational experiences in entrepreneurship scores higher on proactiveness than the group without such experiences.*

The Hypotheses 2 through 4 deal with only marginal effects of gender, major, and educational experiences on entrepreneurship mindsets, and thus the actual effects of the variables might be confounded [90,91]. Therefore, it is imperative to investigate the conditional effects of gender, major, and educational experiences to separate out the unique contribution of each variable [90,91] on the entrepreneurship mindsets. Based on a great deal of evidences for the effect of entrepreneurship education on entrepreneurial attitude [78,92–94], intention [36,78,95–97], and behavior [98–101], we believe that the educational experiences in entrepreneurship would play the most crucial role in the college students' entrepreneurial mindset even after controlling for the effects of gender and major.

Hence, we also suggest the following hypotheses regarding the conditional effect of gender, major, and educational experiences:

**Hypothesis 5 (H5).** *Educational experience is the most influencing factor for the scores of the CS-EMS sub-constructs after controlling for gender and major.*

**Hypothesis 5a (H5a).** *Educational experience is the most influencing factor for innovativeness after controlling for gender and major.*

**Hypothesis 5b (H5b).** *Educational experience is the most influencing factor for need for achievement after controlling for gender and major.*

**Hypothesis 5c (H5c).** *Educational experience is the most influencing factor for risk-taking after controlling for gender and major.*

**Hypothesis 5d (H5d).** *Educational experience is the most influencing factor for autonomy after controlling for gender and major.*

**Hypothesis 5e (H5e).** *Educational experience is the most influencing factor for proactiveness after controlling for gender and major.*

## 3. Materials and Methods

### 3.1. Participants

We used the dataset that was collected for the initial validation of the CS-EMS [25]. At a large private university in Korea, they collected data via emails with an online survey link. A total of 317 students provided completed and valid responses. At the beginning of the online survey, the purpose of the study and the possible use of the data were presented. Only the data from participants who provided consent were analyzed in the current study. Table 1 shows the distribution of the participants' major, grade, and educational experience by gender. Of the 317 participants, 68.5% were males and 31.5% were females. The participants' majors included engineering (47.3%), economics (17.4%), liberal arts (13.9%), social sciences (13.2%), and sciences (8.2%). The majority of the participants was either juniors (28.7%) or seniors (35.7%), while 19.1% were freshmen and 16.6% were sophomores. Among the participants, 52.7% had at least one educational experience with entrepreneurship (e.g., formal classes, extracurricular activities at university, competitions out of university). It is important to be aware that the imbalanced gender representation was largely due to the large number of students from engineering majors (N = 150; 47.3%). Male students majoring in engineering (N = 123) represented 68.5% of the total number of male participants. In addition, 70 male participants majoring in engineering represented 57.4% of the male participants who had some educational experience in entrepreneurship.

### 3.2. Instrument

Jung and Lee [25] developed the CS-EMS with 19 items, and they investigated the evidence related to construct validity and predictive validity with regard to entrepreneurial intentions. Based on their results, the CS-EMS stipulated five sub-factors: *innovativeness*, *need for achievement*, *risk-taking*, *autonomy*, and *proactiveness*. In their study, each sub-factor was operationally defined as follows: (1) *innovativeness*: propensity to seek new opportunities and solutions; (2) *need for achievement*: propensity to achieve something quickly and well; (3) *risk-taking*: propensity to try something with either unclear expectations or the possibility of failure; (4) *autonomy*: propensity to act independently while being reluctant to rely on others; and (5) *proactiveness*: propensity to plan and act in advance. Table 2 presents the English-translated items of the Entrepreneurial Mindset Scale by sub-factor. Each of the items

was measured with a 5-point Likert scale ranging from 1 (strongly disagree) to 5 (strongly agree). Sub-scale scores represent the average of the items under a sub-factor. Higher scores indicate a higher level of the entrepreneurial mindset sub-factor. Table 2 presents the mean, standard deviation, skewness, and kurtosis of each item. The range of item means was 2.85 (SD = 1.13; Item 14) to 4.12 (SD = 0.85; item 4), while the skewness and kurtosis values ranged from −0.85 to 0.28 and from −0.86 to 0.97, respectively.

**Table 1.** Participants' Characteristics.

| Category | Male | | Female | | Total | |
|---|---|---|---|---|---|---|
| | N | % | N | % | N | % |
| *Major* | | | | | | |
| Engineering | 123 | 82.0 | 27 | 18.0 | 150 | 47.3 |
| Science | 18 | 69.2 | 8 | 30.8 | 26 | 8.2 |
| Economics | 32 | 58.2 | 23 | 41.8 | 55 | 17.4 |
| Liberal Arts | 29 | 51.8 | 27 | 48.2 | 44 | 13.9 |
| Social Science | 15 | 50.0 | 15 | 50.0 | 42 | 13.2 |
| *Grade* [a] | | | | | | |
| Freshman | 36 | 60.0 | 24 | 40.0 | 60 | 19.1 |
| Sophomore | 25 | 48.1 | 27 | 51.9 | 52 | 16.6 |
| Junior | 66 | 73.3 | 24 | 26.7 | 90 | 28.7 |
| Senior | 87 | 77.7 | 25 | 22.3 | 112 | 35.7 |
| *Educational Experience in Entrepreneurship* | | | | | | |
| Yes | 122 | 73.1 | 55 | 26.9 | 167 | 52.7 |
| No | 95 | 63.3 | 44 | 36.7 | 150 | 47.3 |
| Total | 217 | 68.5 | 100 | 3.2 | 317 | 100.0 |

Note. [a] Three of the respondents did not provided their grade.

**Table 2.** English-translated College Students' Entrepreneurial Mindset Scale.

| # | Item | M [a] | SD [b] | Skew. [c] | Kurt. [d] |
|---|---|---|---|---|---|
| *Innovativeness* | | | | | |
| Item 1 | I like to take on a new challenge. | 3.65 | 0.91 | −0.42 | −0.36 |
| Item 2 | I try to work in a novel way. | 3.47 | 0.98 | −0.23 | −0.59 |
| Item 3 | I am likely to accept new ideas. | 3.99 | 0.81 | −0.75 | 0.79 |
| Item 4 | I like imaginative ideas. | 4.12 | 0.85 | −0.85 | 0.52 |
| Item 5 | I try to look for new opportunities earlier than others. | 3.74 | 0.90 | −0.32 | −0.39 |
| Item 6 | I persistently try to come up with outstanding ideas. | 3.50 | 0.91 | 0.00 | −0.56 |
| *Need for Achievement* | | | | | |
| Item 7 | I act aggressively to achieve a goal. | 4.08 | 0.78 | −0.78 | 0.97 |
| Item 8 | I am more passionate than others. | 3.82 | 0.84 | −0.33 | −0.29 |
| Item 9 | I have a strong will to achieve something. | 4.02 | 0.79 | −0.61 | 0.32 |
| Item 10 | I persist in pushing forward necessary things against all odds. | 4.09 | 0.76 | −0.63 | 0.50 |
| *Risk-taking* | | | | | |
| Item 11 | I tend to push forward something with high expected value even with high risk. | 3.57 | 1.00 | −0.28 | −0.66 |
| Item 12 | I tend to take risks for new opportunities. | 3.43 | 1.00 | −0.15 | −0.65 |
| Item 13 | I tend to take challenges even when there is a risk of failure. | 3.47 | 0.99 | −0.26 | −0.67 |
| *Autonomy* | | | | | |
| Item 14 | I am reluctant to receive outside aid. | 2.85 | 1.13 | 0.28 | −0.86 |
| Item 15 | I prefer solving problems independently. | 3.42 | 1.04 | −0.35 | −0.52 |
| Item 16 | I prefer acting based on my own decision. | 3.86 | 0.85 | −0.73 | 0.58 |
| *Proactiveness* | | | | | |
| Item 17 | I proactively plan new things. | 3.83 | 0.76 | −0.43 | 0.05 |
| Item 18 | I plan and act in advance rather than waiting for something to be given. | 3.72 | 0.88 | −0.42 | −0.18 |
| Item 19 | I tend to actively overcome hardships rather than attributing to the environment. | 3.79 | 0.82 | −0.42 | −0.03 |

Note. [a] Mean; [b] standard deviation; [c] skewness; and [d] kurtosis.

Jung and Lee [25] found that the Cronbach's $\alpha$ of the whole scale was 0.94, while the Cronbach's $\alpha$s for the *innovativeness*, *need for achievement*, *risk-taking*, *autonomy*, and *proactiveness* sub-scales were 0.88, 0.83, 0.88, 0.77, and 0.80, respectively. In their study, the correlated five-factor model was confirmed based on the results from both exploratory and confirmatory factor analyses. Predictive validity was evidenced by the significant correlations (range: 0.22~0.54) between each of the three start-up intention variables (weak and vague intention, moderate intention, and strong and firm intention) and four sub-factors (*innovativeness*, *need for achievement*, *risk-taking*, and *proactiveness*), except for autonomy. The autonomy sub-scale score had a statistically significant correlation (.11) only with strong and firm intention.

### 3.3. Analytic Procedure

Data analyses were conducted in four phases to fulfill the purposes of the study. In the first phase, we examined the factor structure of the CS-EMS with six groups of interest (i.e., male, female, engineering, non-engineering, educational experiences, no educational experiences) separately using confirmatory factor analysis (CFA). The major reason we selected CFA is that this method is built on theories rather than guided by data [102,103]. Since a correlated five-factor model had already been established by Jung and Lee [25], CFA was considered a more appropriate starting point than exploratory factor analysis (EFA). In addition, CFA is known for providing a more trustworthy solution than EFA for models with multiple factors [102], such as the one used in our study. Most importantly, CFA is a more powerful method to test every element of factorial invariance [28], whereas EFA is capable of testing only factor loading invariance [26].

As shown in Table 2 in the previous section, neither the skewness (range: −0.85~0.28) nor kurtosis (range: −0.86~0.97) of any item appeared to seriously violate the normality assumption of the CFA based on the criteria (skewness ≤ ±2; kurtosis ≤ ±7) suggested by Hair et al. [104] and Byrne [105]. Therefore, we used the maximum likelihood estimation method to evaluate the model [102]. The adequacy of the tested CFA models was evaluated using conventionally reported fit indices, such as the chi-square ($\chi^2$) fit statistic at a 0.05 significance level, the root mean square of approximation (*RMSEA*), the comparative fit index (*CFI*), and the standardized root mean squared residual (*SRMR*). In some conditions with large samples and/or a complex model, $\chi^2$ is too sensitive to retain an acceptable model [102]. Thus, we carefully examined model adequacy, referring to the other fit indices while considering the models acceptable with *RMSEA* ≤ 0.08, the *CFI* ≥ 0.90, and the *SRMR* ≤ 0.08 [102,106,107].

In the second phase, we tested H1. The four levels of factorial invariance (configural, metric, scalar, and strict invariance) were tested sequentially using a MG-CFA model. For example, the configural invariance model was compared with the metric invariance model based on the difference (Δ) in the model fit indices. The model with more invariance constraints is generally expected to have deteriorated fit statistics. A significant value of $\Delta\chi^2$ indicates that the model with more invariance constraints (e.g., the metric invariance model) is poorer than the model with fewer invariance constraints (e.g., the configural invariance model). Like $\chi^2$, $\Delta\chi^2$ may overly reject acceptable models. Therefore, we consulted Δ*RMSEA*, Δ*CFI*, and Δ*SRMR* as well for the cases in which $\Delta\chi^2$ was statistically significant. We used the criteria for acceptable models in accordance with Chen's [108] recommendations. He suggested that a metric invariance model is acceptable when Δ*RMSEA* ≥ 0.010, Δ*CFI* ≥ −0.005, and Δ*SRMR* ≥ 0.025 and that either a scalar or strict invariance model is acceptable when Δ*RMSEA* ≥ 0.010, Δ*CFI* ≥ −0.005, and Δ*SRMR* ≥ 0.005, given a group size < 300.

In the third phase, we tested H2 through H4 by investigating the observed and latent sub-factor mean differences between every pair of compared groups when at least the scalar invariance condition is satisfied [27,55]. In the final phase, we simultaneously tested the effect of the gender, major, and educational experiences on the sub-factors of the CS-EMS using the structural equation modeling framework to test H5. While the observed mean difference between the groups was examined using *IBM SPSS 26*, the remaining analyses (i.e., confirmatory factor analysis, multiple-group confirmatory factor analysis, latent mean comparisons, structural equation modeling) were conducted using *MPlus8*.

## 4. Results

### 4.1. Confirmatory Factor Analysis

Before performing a measurement invariance test, we fitted the correlated five-factor model (Figure 1) to each of the six groups (i.e., male, female, engineering, non-engineering, educational experiences, and no educational experiences) separately. In Figure 1 $\lambda_{ij}$, $\tau_{ij}$, $\delta_{ij}$, and $\theta_{ij}$ represent the factor loading, intercept, unique factor score, and unique variance of the $i_{th}$ factor's $j_{th}$ item, respectively.

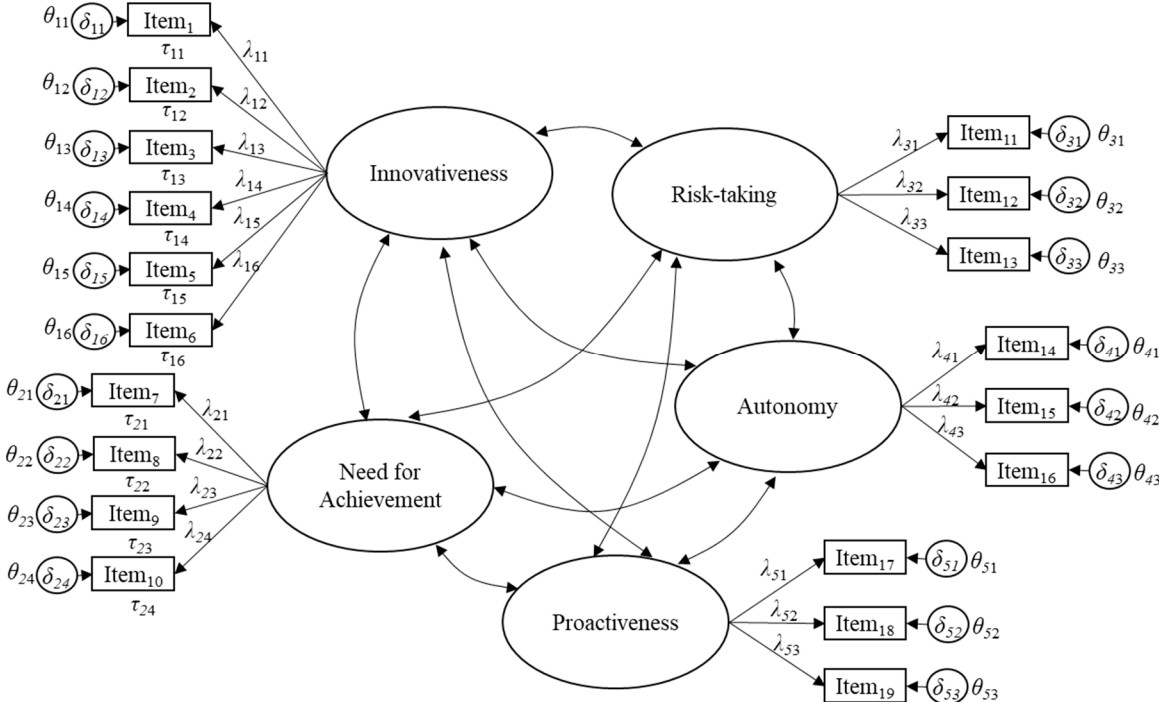

**Figure 1.** The correlated five-factor model of the Entrepreneurial Mindset Scale.

The results of the CFA analyses can be found in Table 3. The chi-square ($\chi^2$) fit statistic for the CFA model was statistically significant for all groups, which means that the tested model does not fit the data. However, the limitation of the $\chi^2$ fit statistic (i.e., it can easily reject a viable model when given a large sample [102]) allowed us to refer to alternative fit statistics, such as *RMSEA*, *CFI*, and *SRMR*. All the alternative fit statistics consistently indicated that the tested CFA model was tenable for all groups; for all *CFI* > 0.90 and *RMSEA* and *SRMR* < 0.08. Thus, the correlated five-factor model without any model modification served as the baseline model for sequential tests of factorial invariance, which were performed in the next analyses.

**Table 3.** Confirmatory Factor Analysis Results for Each of the Six Groups.

| Sub-Groups | $\chi^2$ | df | p-Value | RMSEA | CFI | SRMR |
|---|---|---|---|---|---|---|
| Male | 301.345 | 142 | <0.000 | 0.072 | 0.920 | 0.062 |
| Female | 197.154 | 142 | 0.002 | 0.062 | 0.952 | 0.068 |
| Engineering | 267.909 | 142 | <0.000 | 0.077 | 0.908 | 0.069 |
| Non-Engineering | 261.844 | 142 | <0.000 | 0.071 | 0.934 | 0.058 |
| Experience | 280.792 | 142 | <0.000 | 0.077 | 0.923 | 0.063 |
| No experience | 264.220 | 142 | <0.000 | 0.076 | 0.913 | 0.065 |

Note. RMSEA: the root mean square of approximation; CFI: the comparative fit index; SRMR: the standardized root mean squared residual.

Table 4 presents the Cronbach's $\alpha$s of the whole scale and each of the five sub-scales (*innovativeness*, *need for achievement*, *risk-taking*, *autonomy*, and *proactiveness*) for the six groups. The Cronbach's $\alpha$ of the whole scale ranged from 0.882 to 0.919 while those of the sub-scales ranged from 0.717 to 0.902 across the six groups. All of them appeared to be adequate [109,110].

**Table 4.** Cronbach's $\alpha$s of the CS-EMS by Group.

| Sub-Groups | Whole Scale | Innovative-Ness | Need for Achievement | Risk-Taking | Autonomy | Proactive-Ness |
|---|---|---|---|---|---|---|
| Male | 0.891 | 0.863 | 0.813 | 0.883 | 0.717 | 0.803 |
| Female | 0.919 | 0.902 | 0.853 | 0.878 | 0.851 | 0.796 |
| Engineering | 0.886 | 0.864 | 0.799 | 0.866 | 0.743 | 0.821 |
| Non-Engineering | 0.914 | 0.886 | 0.850 | 0.896 | 0.793 | 0.789 |
| Experience | 0.916 | 0.881 | 0.860 | 0.897 | 0.729 | 0.796 |
| No experience | 0.882 | 0.874 | 0.789 | 0.858 | 0.809 | 0.793 |

### 4.2. Measurement Invariance Test

The results directly addressing Hypothesis 1 (H1: The CS-EMS presents at least scalar invariance across gender, major, and experience groups.) are presented in this section. The results of the hierarchical factorial invariance tests by gender, major, and educational experience are presented in Table 5. In addition to the overall model fit information for each invariance model, the chi-square difference test results and differences in *RMSEA*, *CFI*, and *SRMR* between a less restricted model and a more restricted model are presented. One thing we should address here is that we used the method that does not require a reference variable [26,29,56] to be appointed for identifying the metric and scalar invariance models.

**Table 5.** Factorial Invariance Test results across Gender, Major, and Educational Experiences.

| | $\chi^2$ | *df* | *RMSEA* | *CFI* | *SRMR* | $\Delta\chi^2$ | $\Delta df$ | $\Delta RMSEA$ | $\Delta CFI$ | $\Delta SRMR$ |
|---|---|---|---|---|---|---|---|---|---|---|
| Gender | | | | | | | | | | |
| Configural | 498.499 ** | 284 | 0.069 | 0.932 | 0.064 | | | | | |
| Metric | 521.084 ** | 298 | 0.069 | 0.929 | 0.073 | 22.585 | 14 | 0.000 | −0.003 | 0.009 |
| Scalar | 531.878 ** | 312 | 0.067 | 0.930 | 0.073 | 10.794 | 14 | −0.002 [a] | 0.001 | 0.000 |
| Strict | 565.710 ** | 331 | 0.067 | 0.925 | 0.078 | 33.832 * | 19 | 0.000 | −0.005 | 0.005 |
| Major | | | | | | | | | | |
| Configural | 529.752 ** | 284 | 0.074 | 0.923 | 0.063 | | | | | |
| Metric | 543.050 ** | 298 | 0.072 | 0.923 | 0.070 | 13.298 | 14 | −0.002 [a] | 0.000 | 0.007 |
| Scalar | 559.411 ** | 312 | 0.071 | 0.923 | 0.072 | 16.361 | 14 | −0.001 [a] | 0.000 | 0.002 |
| Strict | 587.722 ** | 331 | 0.070 | 0.920 | 0.081 | 28.311 | 19 | −0.001[a] | −0.003 | 0.009 |
| Educational Experience | | | | | | | | | | |
| Configural | 545.012 ** | 284 | 0.070 | 0.918 | 0.064 | | | | | |
| Metric | 552.262 ** | 298 | 0.073 | 0.920 | 0.067 | 7.250 | 14 | 0.003 | 0.002 | 0.003 |
| Scalar | 574.837 ** | 312 | 0.073 | 0.918 | 0.069 | 22.575 | 14 | 0.000 | −0.002 | 0.002 |
| Strict | 600.762 ** | 331 | 0.072 | 0.915 | 0.076 | 25.925 | 19 | −0.001 [a] | −0.003 | 0.007 |

Note. * $p < 0.05$; ** $p < 0.01$; RMSEA: the root mean square of approximation; CFI: the comparative fit index; SRMR: the standardized root mean squared residual; $\Delta$ represents a difference test for each statistic between less restricted model (e.g., configural invariance model) and more restricted model (e.g., metric invariance model); [a] In these cases, the changes in the *RMSEA* were not expected (i.e., an increase in values).

### 4.2.1. Configural Invariance Model

The configural invariance model holds across gender ($\chi^2 = 529.752$, $df = 284$, $p < 0.001$; $RMSEA = 0.074$; $CFI = 0.923$; $SRMR = 0.063$), major ($\chi^2 = 498.499$, $df = 284$, $p < 0.001$; $RMSEA = 0.069$; $CFI = 0.932$; $SRMR = 0.064$), and educational experience ($\chi^2 = 545.012$, $df = 284$, $p < 0.001$; $RMSEA = 0.070$; $CFI = 0.918$; $SRMR = 0.064$) based on the same criteria for the CFA.

### 4.2.2. Metric Invariance Model

Metric invariance holds for every comparison based on the non-significant chi-square difference tests between the configural invariance model and metric invariance model (gender: $\Delta\chi^2$ = 13.298, $df$ = 14, $p$ = 0.503; major: $\Delta\chi^2$ = 22.585, $df$ = 14, $p$ = 0.067; educational experience: $\Delta\chi^2$ = 7.250, $df$ = 14, $p$ = 0.925). Based on the Chen's [108] recommendation for the metric invariance test with samples sizes less than 300 ($\Delta RMSEA \geq 0.010$, $\Delta CFI \geq$ -0.005, $\Delta SRMR \geq 0.025$), the differences in *RMSEA*, *CFI*, and *SRMR* (gender: $\Delta RMSEA$ = 0.000, $\Delta CFI$ = −0.003, $\Delta SRMR$ = 0.009; major: $\Delta RMSEA$ = −0.002, $\Delta CFI$ = 0.000, $\Delta SRMR$ = 0.007; educational experience: $\Delta RMSEA$ = 0.003, $\Delta CFI$ = 0.002, $\Delta SRMR$ = 0.003) also supported metric invariance across the gender, major, and educational experience groups.

### 4.2.3. Scalar Invariance Model

After imposing invariant intercept constraints, the chi-square difference tests between the metric invariance model and scalar invariance model were not statistically significant for all comparisons (gender: $\Delta\chi^2$ = 16.361, $df$ = 14, $p$ = 0.292; major: $\Delta\chi^2$ = 10.794, $df$ = 14, $p$ = 0.702; educational experience: $\Delta\chi^2$ = 22.575, $df$ = 14, $p$ = 0.068). There were no outstanding changes in *RMSEA*, *CFI*, and *SRMR* (gender: $\Delta RMSEA$ = −0.001, $\Delta CFI$ = 0.000, $\Delta SRMR$ = 0.002; major: $\Delta RMSEA$ = −0.002, $\Delta CFI$ = 0.001, $\Delta SRMR$ = 0.000; educational experience: $\Delta RMSEA$ = 0.000, $\Delta CFI$ = −0.002, $\Delta SRMR$ = 0.002) based on Chen's [108] criteria for the scalar invariance test with samples of less than 300 ($\Delta RMSEA \geq 0.010$, $\Delta CFI \geq −0.005$, $\Delta SRMR \geq 0.005$). Hence, the results confirmed Hypothesis 1a (H1a: The CS-EMS presents at least scalar invariance between the male and female groups.), Hypothesis 1b (H1b: The CS-EMS presents at least scalar invariance between the engineering and non-engineering groups.), and Hypothesis 1c (H1c: The CS-EMS presents at least scalar invariance between the experience and no-experience groups.).

### 4.2.4. Strict Invariance Model

The chi-square difference tests between the scalar invariance model and strict invariance model were not significant across the pairs based on either major or educational experiences (major: $\Delta\chi^2$ = 28.311, $df$ = 19, $p$ = 0.078; educational experience: $\Delta\chi^2$ = 25.925, $df$ = 19, $p$ = 0.132). Based on Chen's [108] suggestions for strict invariance tests with samples of less than 300 ($\Delta RMSEA \geq 0.010$, $\Delta CFI \geq −0.005$, $\Delta SRMR \geq 0.005$), the changes in the other fit indices were negligible (major: $\Delta RMSEA$ = −0.001, $\Delta CFI$ = −0.003; educational experience: $\Delta RMSEA$ = −0.001, $\Delta CFI$ = −0.003, $\Delta SRMR$ = 0.007) except for *SRMR* (major: $\Delta SRMR$ = 0.009; educational experiences: $\Delta SRMR$ = 0.007). For the gender comparison, the chi-square difference test results indicated that the strict invariance model was significantly worse than the scalar invariance model. In addition, changes in the two other fit indices ($\Delta CFI$ = −0.005; $\Delta SRMR$ = 0.005) indicated that the strict invariance model was worse than the scalar invariance model. However, we did not pursue partial strict invariance since scalar invariance is a sufficient condition for latent and observed mean comparisons [26,29]. We provide the measurement parameter estimates ($\lambda_{ij}$, $\tau_{ij}$, and $\theta_{ij}$) of the final confirmed factorial invariance model by gender, major, and educational experience in Appendix A (Tables A1–A3).

### 4.3. Comparison of Latent and Observed Means

In this section, we present the results that are directly related to Hypothesis 2 (H2: The male group scores higher on each of the five sub-constructs of the CS-EMS than the female group.), Hypothesis 3 (H3: The engineering group scores higher on each of the five sub-constructs of the CS-EMS than the non-engineering group.), and Hypothesis 4 (H4: The group with educational experiences in entrepreneurship scores higher on each of the five sub-constructs of the CS-EMS than the group without such experiences.). The latent means were tested between every set of the compared groups under the finally confirmed factorial invariance model using a MG-CFA. For the groups based on major (engineering vs. non-engineering) and educational experience (with educational experiences in

entrepreneurship vs. without educational experiences in entrepreneurship), the latent means were compared using the strict invariance model. To compare the latent means between males and females, the scalar invariance model was used. For each comparison, non-engineering students, females, and the students without educational experiences in entrepreneurship served reference groups with a fixed latent mean score of zero. Table 6 shows the estimated sub-scale latent and observed means of the groups by gender, major, and educational experience.

**Table 6.** Factorial Invariance Test results across Gender, Major, and Educational Experience Groups.

| | | Gender | | Major | | Educational Experience | |
|---|---|---|---|---|---|---|---|
| | | Male M (SE) | Female M (SD) | Eng. M (SE) | Non-Eng. M (SD) | Yes M (SE) | No M (SD) |
| Innovativeness | $\xi_i$ | 0.44 (0.15) ** | 0.00 (0.00) | 0.37 (0.11) ** | 0.00 (0.00) | 0.42 (0.13) ** | 0.00 (0.00) |
| | $O_i$ | 3.83(0.66) ** | 3.57 (0.77) | 3.89 (0.63) ** | 3.62 (0.75) | 3.86 (0.71) ** | 3.61 (0.68) |
| Need for Achievement | $\xi_i$ | 0.22 (0.15) * | 0.00 (0.00) | 0.17 (0.11) ** | 0.00 (0.00) | 0.21 (0.13) ** | 0.00 (0.00) |
| | $O_i$ | 4.04 (0.60) * | 3.92 (0.73) | 4.06 (0.60) * | 3.95 (0.68) | 4.06 (0.68) ** | 3.94 (0.61) |
| Risk-taking | $\xi_i$ | 0.33 (0.13) ** | 0.00 (0.00) | 0.28 (0.12) ** | 0.00 (0.00) | 0.44 (0.13) ** | 0.00 (0.00) |
| | $O_i$ | 3.58 (0.90) ** | 3.29 (0.87) | 3.63 (0.87) ** | 3.36 (0.91) | 3.66 (0.90) ** | 3.30 (0.85) |
| Autonomy | $\xi_i$ | 0.05 (0.15) * | 0.00 (0.00) | −0.09 (0.12) * | 0.00 (0.00) | −0.13 (0.11) ** | 0.00 (0.00) |
| | $O_i$ | 3.37 (0.79) * | 3.39 (0.96) | 3.37 (0.82) * | 3.37 (0.87) | 3.33 (0.81) ** | 3.44 (0.89) |
| Proactiveness | $\xi_i$ | 0.40 (0.16) * | 0.00 (0.00) | 0.14 (0.12) * | 0.00 (0.00) | 0.52 (0.13) ** | 0.00 (0.00) |
| | $O_i$ | 3.85 (0.64) ** | 3.62 (0.78) | 3.82 (0.67) * | 3.85 (0.71) | 3.93 (0.67) ** | 3.61 (0.68) |

Note. $\xi i$: Estimated latent mean; Oi: observed mean; M: mean; SE: standard error of the estimated mean; SD: standard deviation; * $p < 0.05$; ** $p < 0.01$.

### 4.3.1. Comparison Based on Gender

Among the five sub-scales, the male group had significantly higher latent means on the *innovativeness* ($M = 0.44$, $SE = 0.15$), *risk-taking* ($M = 0.33$, $SE = 0.13$), and *proactiveness* ($M = 0.40$, $SE = 0.16$) sub-scales than the female group, which confirmed Hypothesis 2a (H2a: The male group scores higher on innovativeness than the female group.), Hypothesis 2c (H2c: The male group scores higher on risk-taking than the female group.), and Hypothesis 2e (H2e: The male group scores higher on proactiveness than the female group.). The latent means of two sub-scales (*need for achievement* and *autonomy*) did not differ across the groups, and thus Hypothesis 2b (H2b: The male group scores higher on need for achievement than the female group.) and Hypothesis 2d (H2d: The male group scores higher on autonomy than the female group.) were rejected. Regarding the sub-scales' observed means, the male group scored higher on the *innovativeness* ($M = 3.83$, $SD = 0.66$), *risk-taking* ($M = 3.58$, $SD = 0.90$), and *proactiveness* ($M = 3.85$, $SD = 0.64$) sub-scales than the female group. The effect sizes (Cohen's *d*, 1988) for the observed mean scores of *innovativeness*, *risk-taking*, and *proactiveness* were 0.38, 0.33, and 0.34, respectively, which indicate small to medium effects (Cohen, 1988).

### 4.3.2. Comparison Based on Major

The engineering major group had significantly higher latent means for the *innovativeness* (M = 0.37, SE = 0.11) and *risk-taking* ($M = 0.28$, $SE = 0.12$) sub-scale compared to the non-engineering major group, which supported Hypothesis 3a (H3a: The engineering group scores higher on innovativeness than the non-engineering group.) and Hypothesis 3e (H3e: The engineering group scores higher on proactiveness than the non-engineering group.). Yet, the two groups did not differ in the latent means of the *need for achievement*, *autonomy*, and *proactiveness* sub-scales, and thus we rejected Hypothesis 3b (H3b: The engineering group scores higher on need for achievement than the non-engineering group.), Hypothesis 3c (H3c: The engineering group scores higher on risk-taking than the non-engineering group.), and Hypothesis 3d (H3d: The engineering group scores higher on autonomy than the non-engineering group.). The same pattern of significant differences could be found in the observed sub-scale mean comparisons. The engineering major group had higher observed sub-scale mean scores for both *innovativeness* (M = 3.89, SD = 0.63) and *risk-taking* (M = 3.62, SD = 0.75) compared to the

non-engineering major group. The Cohen's *d* effect sizes for the *innovativeness* and *risk-taking* sub-scales were 0.39 and 0.29, respectively, which indicate small (0.02) to medium effects (0.05) according to Cohen (1988).

### 4.3.3. Comparison Based on Educational Experiences in Entrepreneurship

The group with educational experiences in entrepreneurship scored substantially higher on the *innovativeness* ($M = 0.42$, $SE = 0.13$), *risk-taking* ($M = 0.44$, $SE = 0.13$), and *proactiveness* ($M = 0.52$, $SE = 0.13$) sub-scales than the group without such experiences, which confirmed Hypothesis 4a (H4a: The group with educational experiences in entrepreneurship scores higher on each of the five sub-constructs of the CS-EMS than the group without such experiences.), Hypothesis 4c (H4c: The group with educational experiences in entrepreneurship scores higher on risk-taking than the group without such experiences.), and Hypothesis 4e (H4e: The group with educational experiences in entrepreneurship scores higher on proactiveness than the group without such experiences.). The remaining sub-scales, *need for achievement* and *autonomy*, did not differ across the groups, and thus we rejected Hypothesis 4b (H4b: The group with educational experiences in entrepreneurship scores higher on need for achievement than the group without such experiences.) and Hypothesis 4d (H4d: The group with educational experiences in entrepreneurship scores higher on autonomy than the group without such experiences.). For the observed sub-scale scores, the group with educational experiences in entrepreneurship scored higher on the *innovativeness* ($M = 3.86$, $SD = 0.71$), *risk-taking* ($M = 3.66$, $SD = 0.90$), and *proactiveness* ($M = 3.93$, $SD = 0.67$) sub-scales than the group without educational experiences. The effect sizes for the observed mean scores of the *innovativeness*, *risk-taking*, and *proactiveness* sub-scales were 0.37, 0.41, and 0.48, respectively, which indicate small to medium effects (Cohen, 1988).

### 4.4. Structural Equation Modeling: Tests of Conditional Group Effects on Each Sub-Scale

This section addresses Hypothesis 5 (H5: Educational experience is the most influencing factor for the scores of the CS-EMS sub-constructs after controlling for gender and major.) directly. In the previous phase, we tested the sub-scales' latent means across each pair of groups without considering the effect of the other groups. Thus, we investigated the conditional effect of the group on the latent scores of the CS-EMS sub-scales by including all three grouping variables as independent variables in the model under the structural equation modeling framework (Figure 2). By doing so, the effect of the overrepresentation of male participants majoring in engineering can be controlled, and we can single out the effects of each of the three variables.

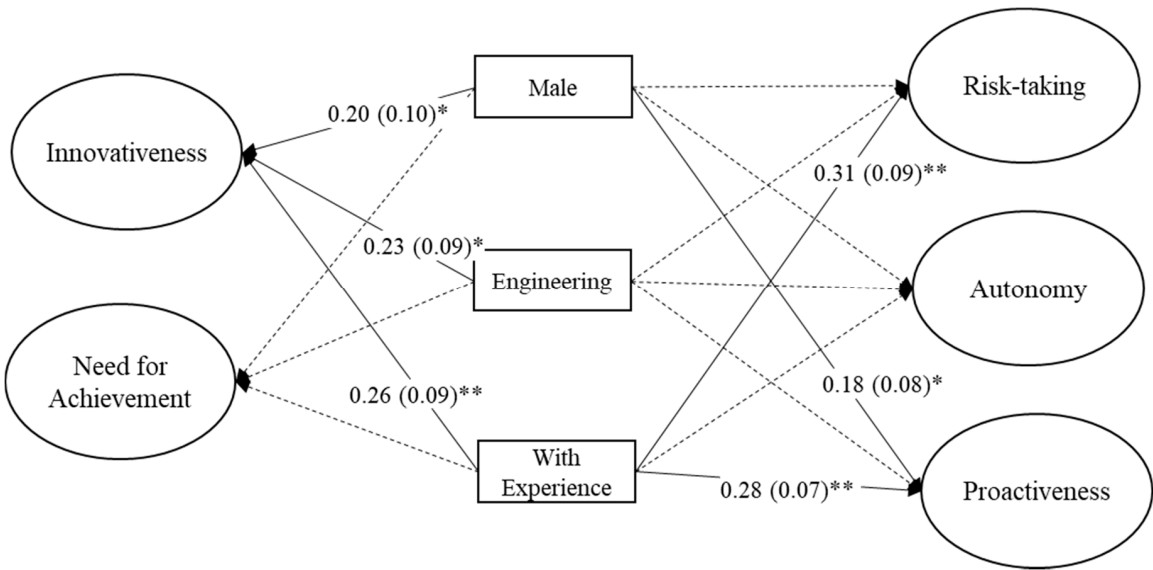

**Figure 2.** Tests of the Grouping Variable Effects on the Entrepreneurial Mindset Sub-scales.



### 4.4.1. Innovativeness

In the simple latent and observed mean comparisons, the *innovativeness* sub-scale scores significantly differed across all comparison pairs. Even after controlling for the remaining variables, each of the three grouping variables (gender, major, and experience) had a significant effect on the *innovativeness* sub-scale score. To interpret the estimated effect, the male group's *innovativeness* score was 0.20 higher than the female group when the effect of the major and experience variables was considered. The engineering major group's *innovativeness* score was 0.23 higher than the non-engineering major group when controlling for the effect of major and gender. The group with educational experiences in entrepreneurship scored 0.26 higher on the *innovativeness* sub-scale than the group without educational experiences when the effects of major and gender were accounted for. To sum up the results, we considered Hypothesis 5a (H5a: Educational experience is the most influencing factor for *innovativeness* after controlling for gender and major.) to be supported.

### 4.4.2. Need for Achievement

Similarly, in the results for the simple latent and observed mean comparisons, none of the three grouping variables (*gender*, *major*, and *experience*) had a significant effect on the score of the *need for achievement* sub-scale. Thus, we rejected Hypothesis 5b (H5b: Educational experience is the most influencing factor for *need for achievement* after controlling for gender and major.).

### 4.4.3. Risk-Taking

Whereas the *risk-taking* sub-scale scores significantly differed across all pairs of comparison in the simple latent and observed mean difference tests, only experience had a significant effect on the *risk-taking* sub-scale. That is, the score for the *risk-taking* sub-scale was 0.31 higher for the group with educational experiences in entrepreneurship than the group without such experiences after controlling for the effects of gender and major. Interestingly, the effects of *gender* and *major* disappeared when the other grouping variables were considered. Hence, the result confirmed Hypothesis 5c (H5c: Educational experience is the most influencing factor for *risk-taking* after controlling for gender and major.).

### 4.4.4. Autonomy

None of the three grouping variables (*gender*, *major*, and *experience*) had a significant effect on the *need for achievement* sub-scale score, which was consistent with the results of the simple latent and observed mean comparisons. Therefore, we rejected Hypothesis 5d (H5d: Educational experience is the most influencing factor for *autonomy* after controlling for gender and major.).

### 4.4.5. Proactiveness

In the simple latent and observed mean comparisons, the *proactiveness* sub-scale scores significantly differed across groups based on either gender or educational experiences. A similar pattern was found through a structural equation modeling analysis. The same two grouping variables (gender and experience) had a significant effect on the *proactiveness* sub-scale scores. Specifically, the male group's *proactiveness* score was 0.18 higher than that of the female group after controlling for the effect of major and experience, while the group with educational experiences in entrepreneurship scored 0.28 higher on the *proactiveness* sub-scale than the group without such experiences when the effects of major and gender was considered. Hence, Hypothesis 5e (H5e: Educational experience is the most influencing factor for *proactiveness* after controlling for gender and major.) was confirmed by the result.

To briefly summarize the results of the current study by the hypotheses, we present Table 7. Table 7 provides the information on whether each of the hypotheses was confirmed or not.

**Table 7.** Summary of the Study Results based on the Research Hypotheses.

| Hypothesis | Result |
|---|---|
| *H1: The CS-EMS presents at least scalar invariance across gender, major, and experience groups.* | |
|     H1a: The CS-EMS presents at least scalar invariance between the male and female groups. | Confirmed |
|     H1b: The CS-EMS presents at least scalar invariance between the engineering and non-engineering groups. | Confirmed |
|     H1c: The CS-EMS presents at least scalar invariance between the experience and no-experience groups. | Confirmed |
| *H2: The male group scores higher on each of the five sub-constructs of the CS-EMS than the female group.* | |
|     H2a: The male group scores higher on innovativeness than the female group. | Confirmed |
|     H2b: The male group scores higher on need for achievement than the female group. | Rejected |
|     H2c: The male group scores higher on risk-taking than the female group. | Confirmed |
|     H2d: The male group scores higher on autonomy than the female group. | Rejected |
|     H2e: The male group scores higher on proactiveness than the female group. | Confirmed |
| *H3: The engineering group scores higher on each of the five sub-constructs of the CS-EMS than the non-engineering group.* | |
|     H3a: The engineering group scores higher on innovativeness than the non-engineering group. | Confirmed |
|     H3b: The engineering group scores higher on need for achievement than the non-engineering group. | Rejected |
|     H3c: The engineering group scores higher on risk-taking than the non-engineering group. | Rejected |
|     H3d: The engineering group scores higher on autonomy than the non-engineering group. | Rejected |
|     H3e: The engineering group scores higher on proactiveness than the non-engineering group. | Confirmed |
| *H4: The group with educational experiences in entrepreneurship scores higher on each of the five sub-constructs of the CS-EMS than the group without such experiences.* | |
|     H4a: The group with educational experiences in entrepreneurship scores higher on innovativeness than the group without such experiences. | Confirmed |
|     H4b: The group with educational experiences in entrepreneurship scores higher on need for achievement than the group without such experiences. | Rejected |
|     H4c: The group with educational experiences in entrepreneurship scores higher on risk-taking than the group without such experiences. | Confirmed |
|     H4d: The group with educational experiences in entrepreneurship scores higher on autonomy than the group without such experiences. | Rejected |
|     H4e: The group with educational experiences in entrepreneurship scores higher on proactiveness than the group without such experiences. | Confirmed |
| *H5: Educational experience is the most influencing factor for the scores of the CS-EMS sub-constructs after controlling for gender and major.* | |
|     H5a: Educational experience is the most influencing factor for innovativeness after controlling for gender and major. | Confirmed |
|     H5b: Educational experience is the most influencing factor for need for achievement after controlling for gender and major. | Rejected |
|     H5c: Educational experience is the most influencing factor for risk-taking after controlling for gender and major. | Confirmed |
|     H5d: Educational experience is the most influencing factor for autonomy after controlling for gender and major. | Rejected |
|     H5e: Educational experience is the most influencing factor for proactiveness after controlling for gender and major. | Confirmed |

## 5. Discussion

### 5.1. Findings and Implications

We began this study with the motivation to contribute to the literature on entrepreneurship in higher education by investigating the untouched topic of measurement invariance of the CS-EMS, which is required for cross-group mean comparisons [26–29,55]. To do so, we focused on comparing the groups of participants by gender, major (engineering vs. non-engineering), or educational experiences in entrepreneurship. In this section, we summarized the findings based on the outline of the analytic procedures and results: (1) confirmatory factor analysis, (2) measurement invariance tests, (3) cross-group latent and observed mean comparisons, and (4) examination of the conditional effects of the grouping variables, while discussing the implications of each finding.

First, we found that the correlated five factor model [25] was viable for all six groups (male, female, engineering, non-engineering, educational experience, and no educational experience). This finding was not consistent with previous studies [22,23,111], in which only three sub-factors (*innovativeness*, *risk-taking*, and *proactiveness*) were included. Instead, our findings are more closely aligned with the study of Lumpkin and Dess [42], in which they introduced five traits (*innovativeness*, *risk-taking*, *proactiveness*, *autonomy*, and *competitive aggressiveness*) related to entrepreneurial orientation at the organizational level. Given the inconsistency in the structure of individual-level entrepreneurial propensity/ orientation/ mindset, our findings might encourage future research to validate the factor structure of the CS-EMS in different countries or different educational contexts. We provided the English-translated items of the CS-EMS in the hope of observing further investigations related to the structural validity of the CS-EMS.

Secondly, we found that the strict invariance model was tenable across both pairs of groups for major and educational experience. To put it another way, all levels (factor loadings, intercepts, and unique variances) of the measurement property operated in the same way between the male and female groups as well as between the group with educational experiences in entrepreneurship and the group without such experiences. Yet, only scalar invariance was retained between the male and female groups, which means that the extent of the unique variance – the approximation of measurement errors – was not equivalent between the groups. Because the required condition (i.e., at least scalar invariance) for comparing latent and observed group means was met, we did not pursue partial strict invariance [26,29,55]. In some studies, measurement invariance of entrepreneurial attitude and intention was tested across only gender. For example, measurement invariance of entrepreneurial intention held between males and females [74,112]. In addition, measurement invariance of entrepreneurial attitude was also established between males and females. To our knowledge, this study is the first to investigate measurement invariance of the entrepreneurial mindset not only between the gender groups but also between groups based on major and educational experience. As a result, we contribute to entrepreneurship literature by reporting evidence of measurement invariance across plausible groups of interest in the context of higher education.

Third, we tested the latent means for each comparison based upon the established measurement invariance model. We also examined the observed mean differences between each set of the compared groups. The pattern of significant difference was consistent between the latent and observed mean comparisons. Male students had generally higher scores on the CS-EMS sub-scales except for *need for achievement* and *autonomy*, compared to the female students. This finding is consistent with formal studies in which male participants scored higher on other entrepreneurship-related variables, such as entrepreneurial orientation [113,114], intention [79,112], and attitude [112]. However, some inconsistent results on the gender difference also exist [115,116]. The gender difference found in the current study raises the old but persistent question, "Is it innate or socially constructed?" Since our study used the term "gender" as analogous to biological sex, future research should thoroughly investigate whether the gender difference in the CS-EMS is given or constructed, following the example of Goktan and Gupta's [113] study by including the concepts of both biological sex and gender (masculinity

vs. femininity). In the comparison by major, the engineering-major group scored higher on the *innovativeness* and *risk-taking* sub-scales. Unfortunately, we could not find any study that directly compares the entrepreneurial orientation or mindset between engineering majors and non-engineering majors. Therefore, it is not possible to discuss the finding in relation to the results of other studies. Instead, one plausible explanation might be that engineering is a field in which males are dominant in most countries [117,118] and, due to the effects of gendered stereotypes, the male participants might have higher self-efficacy and more positive self-reflection than the female participants in our study. Regarding educational experiences in entrepreneurship, the students with experiences showed higher scores on the *innovativeness*, *risk-taking*, and *proactiveness* sub-scales than the students without such experiences. This finding implies that two sub-scales (*need for achievement* and *autonomy*) might not be the outcomes of entrepreneurship education, while the other three sub-scales (*innovativeness*, *risk-taking*, and *proactiveness*) might be. Even though the five correlated-factor model was sustained for the CS-EMS, the *need for achievement* and *autonomy* sub-scales might not measure educational impact effectively. In addition, those sub-scales were found to not be closely related to the entrepreneurial intention variables described by Jung and Lee [25]. Yet, those sub-factors might positively predict other career-related variables (e.g., career adaptability). Further research is needed to investigate this matter. Thus, we are very reluctant to claim that these two sub-scales are not useful.

Finally, as far as the conditional effects of the grouping variables are concerned, only three sub-scales (*innovativeness*, *risk-taking*, and *proactiveness*) of the CS-EMS were influenced by at least one of the grouping variables, whereas the *need for achievement* and *autonomy* sub-scales were not influenced by any of those variables. Among the three grouping variables, the educational experience variable appeared to have the most influence on the *innovativeness*, *risk-taking*, and *proactiveness* sub-scales, as the largest difference between the two groups was observed for these three sub-scales. This finding suggests that the factor with the most influence on the entrepreneurial mindset is educational experiences in entrepreneurship, and the effect of gender and major might be confounded after students have educational experiences. However, our speculation might not be appropriate for making causal inferences within this study. We will revisit this issue when discussing the limitations of the study.

## 5.2. Limitations and Suggestions for Future Studies

Despite its values and contributions, the current study is not free from limitations. The first limitation is related to generalizability. Since we collected data at only one university, which is one of the top colleges in Korea, the results of the current study might not be applicable to other contexts, such as colleges located in other places domestically or globally. To address this limitation, more replication studies should comprehensively discuss the generalizability issue regarding the structural validity and measurement invariance of the CS-EMS. In particular, cross-cultural measurement invariance tests between different countries could be added to the future research agenda. The second limitation is related to the nature of the self-reported assessment tool. Because the CS-EMS measures the extent of the entrepreneurial mindset based on self-reports, some sort of bias (e.g., distribution leaning toward socially desirable values or insincere responses [1]) might have confounded the actual status of the participants' entrepreneurial mindset. Hence, educators or researchers should carefully interpret the scores from the CS-EMS while collecting more evidence regarding educational impact using multiple assessments (e.g., peer evaluation, portfolios, project products). The third limitation is that we cannot make any inferences regarding the causal relationship between the participants' educational experiences and the level of their entrepreneurial mindset. Our data were cross-sectionally collected survey data, and the *experience* variable was made based on heterogeneous past educational experiences, including semester-length formal entrepreneurship classes, extracurricular activities with varying hours, out-of-college competitions to conceive plausible business ideas, and so on. Thus, future research should incorporate an experimental design that can validly measure the actual impact of entrepreneurship education using the CS-EMS. If the design includes pre-and post-measurement, longitudinal measurement invariance should be tested [28,119] before proceeding to latent or observed

mean comparisons between the pre- and post-scores. However, future researchers should be aware that measurement invariance needs a sufficient amount of data [120–122].

## 6. Conclusions

Given the limited evidence regarding the quality of the currently available assessment tools, the CS-EMS would be more useful than the other tools [20–24] for the educators who want to validly measure the educational outcomes or design their own entrepreneurship guided by the current status of the college students' entrepreneurial mindset. In an earlier study, the validity of CS-EMS had been supported by the evidence grounded on structural validity (the five correlated-factor structure) and predictive validity with entrepreneurial intention [25]. The current study provided evidence of measurement invariance, which indicates validity based on the use of assessment results [123], and it legitimately uses CS-EMS scores to compare different groups. Based on the satisfied conditions (scalar or strict invariance) for the cross-group mean comparison, the simple between-group comparisons revealed that the male engineering majors with educational experience generally scored higher on the CS-EMS subscales than their counterparts. As far as the major variable is concerned, the engineering students scored higher on innovativeness compared to the non-engineering students, which might be due to the majors' technology orientation. Regarding the difference based on gender, educators should be aware that female students showed a lower level of innovativeness and proactiveness than the male students. To cultivate the development of sustainable entrepreneurship among female students, universities may invigorate female support programs in entrepreneurial education [124].

Furthermore, we found that educational experience in entrepreneurship is the factor with the most influence on the three sub-scales (*innovativeness*, *risk-taking*, and *proactiveness*) which have been acknowledged to be the core characteristics of entrepreneurial individuals [22,23]. That finding also implies that the CS-EMS has potential as an assessment to efficiently measure the effectiveness of the entrepreneurial education targeting the sub-construct of the CS-EMS. Our finding supports former studies stating that entrepreneurship education is an important factor for building an entrepreneurial mindset [125]. Entrepreneurially-oriented educational programs might enable students to obtain the attitudes needed to gain practical experience and have a positive impact on students' entrepreneurial intentions [32,124]. However, to confirm the causal relationship between educational experiences and the entrepreneurial mindset, further studies with an experimental design are required to gain causal evidence. As a final remark, we would like to gladly introduce the CM-EMS items for future researchers in other countries, and we hope for future studies that perform cross-cultural comparisons using the CS-EMS.

**Author Contributions:** Conceptualization, E.J. and Y.L.; methodology, E.J.; formal analysis, E.J.; writing—original draft preparation, E.J. and Y.L.; writing—revision and editing, E.J. and Y.L.; validation, Y.L. All authors have read and agreed to the published version of the manuscript.

**Funding:** The authors did not receive any funding for the study.

**Conflicts of Interest:** The authors declare no conflict of interest.

# Appendix A

**Table A1.** Parameter Estimates of the Scalar Invariance Model: Groups based on Gender.

| Item | Factor Loading ($\lambda_{ij}$) | | Intercept ($\tau_{ij}$) | | Residual Variances ($\theta_{ij}{}^{G}$) | | | |
| | | | | | Male | | Female | |
| | Estimate | S.E. | Estimate | S.E. | Estimate | S.E. | Estimate | S.E. |
|---|---|---|---|---|---|---|---|---|
| Innovativeness | | | | | | | | |
| Item 1 | 0.69 | (0.05) | 3.75 | (0.06) | 0.28 | (0.03) | 0.26 | (0.05) |
| Item 2 | 0.75 | (0.05) | 3.57 | (0.06) | 0.31 | (0.04) | 0.25 | (0.05) |
| Item 3 | 0.54 | (0.04) | 4.07 | (0.05) | 0.29 | (0.03) | 0.33 | (0.05) |
| Item 4 | 0.49 | (0.04) | 4.19 | (0.05) | 0.43 | (0.04) | 0.48 | (0.07) |
| Item 5 | 0.61 | (0.05) | 3.82 | (0.05) | 0.36 | (0.04) | 0.40 | (0.06) |
| Item 6 | 0.60 | (0.05) | 3.59 | (0.06) | 0.42 | (0.05) | 0.35 | (0.06) |
| Need for Achievement | | | | | | | | |
| Item 7 | 0.54 | (0.04) | 4.12 | (0.05) | 0.27 | (0.03) | 0.26 | (0.05) |
| Item 8 | 0.58 | (0.05) | 3.86 | (0.05) | 0.28 | (0.04) | 0.41 | (0.07) |
| Item 9 | 0.58 | (0.04) | 4.06 | (0.05) | 0.23 | (0.03) | 0.22 | (0.05) |
| Item 10 | 0.48 | (0.04) | 4.12 | (0.05) | 0.29 | (0.03) | 0.33 | (0.06) |
| Risk-taking | | | | | | | | |
| Item 11 | 0.82 | (0.05) | 3.66 | (0.07) | 0.36 | (0.04) | 0.27 | (0.05) |
| Item 12 | 0.92 | (0.05) | 3.53 | (0.07) | 0.15 | (0.03) | 0.16 | (0.07) |
| Item 13 | 0.81 | (0.05) | 3.55 | (0.06) | 0.30 | (0.04) | 0.38 | (0.07) |
| Autonomy | | | | | | | | |
| Item 14 | 0.73 | (0.06) | 2.87 | (0.07) | 0.66 | (0.08) | 0.60 | (0.11) |
| Item 15 | 0.87 | (0.06) | 3.43 | (0.07) | 0.23 | (0.07) | 0.08 | (0.09) |
| Item 16 | 0.49 | (0.05) | 3.87 | (0.05) | 0.47 | (0.05) | 0.41 | (0.07) |
| Proactiveness | | | | | | | | |
| Item 17 | 0.55 | (0.04) | 3.89 | (0.05) | 0.19 | (0.03) | 0.33 | (0.06) |
| Item 18 | 0.63 | (0.05) | 3.80 | (0.06) | 0.31 | (0.04) | 0.29 | (0.06) |
| Item 19 | 0.55 | (0.04) | 3.87 | (0.05) | 0.23 | (0.03) | 0.47 | (0.08) |

Note. Each item and its number correspond to those in Table 2.

**Table A2.** Parameter Estimates of the Scalar Invariance Model: Groups based on Major

| Item | Factor Loading ($\lambda_{ij}$) | | Intercept ($\tau_{ij}$) | | Residual Variances ($\theta_{ij}$) | |
| | Estimate | S.E. | Estimate | S.E. | Estimate | S.E. |
|---|---|---|---|---|---|---|
| Innovativeness | | | | | | |
| Item1 | 0.79 | (0.06) | 3.52 | (0.07) | 0.28 | (0.03) |
| Item 2 | 0.87 | (0.06) | 3.32 | (0.08) | 0.29 | (0.03) |
| Item 3 | 0.63 | (0.05) | 3.88 | (0.06) | 0.30 | (0.03) |
| Item 4 | 0.57 | (0.05) | 4.02 | (0.06) | 0.45 | (0.04) |
| Item 5 | 0.70 | (0.06) | 3.61 | (0.07) | 0.37 | (0.03) |
| Item 6 | 0.71 | (0.06) | 3.38 | (0.07) | 0.39 | (0.04) |
| Need for Achievement | | | | | | |
| Item 7 | 0.62 | (0.05) | 4.03 | (0.06) | 0.27 | (0.03) |
| Item 8 | 0.67 | (0.05) | 3.77 | (0.06) | 0.31 | (0.03) |
| Item 9 | 0.67 | (0.05) | 3.97 | (0.06) | 0.24 | (0.03) |
| Item 10 | 0.55 | (0.05) | 4.05 | (0.05) | 0.31 | (0.03) |
| Risk-taking | | | | | | |
| Item 11 | 0.81 | (0.06) | 3.47 | (0.07) | 0.33 | (0.03) |
| Item 12 | 0.92 | (0.06) | 3.31 | (0.08) | 0.15 | (0.03) |
| Item 13 | 0.81 | (0.06) | 3.36 | (0.07) | 0.33 | (0.03) |

**Table A2.** *Cont.*

| Item | Factor Loading ($\lambda_{ij}$) | | Intercept ($\tau_{ij}$) | | Residual Variances ($\theta_{ij}$) | |
|---|---|---|---|---|---|---|
| | Estimate | S.E. | Estimate | S.E. | Estimate | S.E. |
| Autonomy | | | | | | |
| Item 14 | 0.81 | (0.07) | 2.89 | (0.08) | 0.64 | (0.07) |
| Item 15 | 0.98 | (0.07) | 3.46 | (0.08) | 0.17 | (0.07) |
| Item 16 | 0.53 | (0.06) | 3.89 | (0.06) | 0.46 | (0.04) |
| Proactiveness | | | | | | |
| Item 17 | 0.61 | (0.05) | 3.79 | (0.06) | 0.23 | (0.03) |
| Item 18 | 0.70 | (0.06) | 3.67 | (0.07) | 0.31 | (0.03) |
| Item 19 | 0.61 | (0.05) | 3.76 | (0.06) | 0.31 | (0.03) |

Note. Each item and its number correspond to those in Table 2.

**Table A3.** Parameter Estimates of the Scalar Invariance Model: Groups based on Educational Experiences.

| Item | Factor Loading ($\lambda_{ij}$) | | Intercept ($\tau_{ij}$) | | Residual Variances ($\theta_{ij}$) | |
|---|---|---|---|---|---|---|
| | Estimate | S.E. | Estimate | S.E. | Estimate | S.E. |
| Innovativeness | | | | | | |
| Item 1 | 0.70 | (0.05) | 3.50 | (0.07) | 0.27 | (0.03) |
| Item 2 | 0.77 | (0.06) | 3.30 | (0.07) | 0.29 | (0.03) |
| Item 3 | 0.56 | (0.05) | 3.87 | (0.06) | 0.30 | (0.03) |
| Item 4 | 0.50 | (0.05) | 4.01 | (0.06) | 0.45 | (0.04) |
| Item 5 | 0.62 | (0.05) | 3.60 | (0.06) | 0.38 | (0.03) |
| Item 6 | 0.62 | (0.05) | 3.37 | (0.06) | 0.39 | (0.04) |
| Need for Achievement | | | | | | |
| Item 7 | 0.55 | (0.05) | 4.02 | (0.06) | 0.27 | (0.03) |
| Item 8 | 0.58 | (0.05) | 3.76 | (0.06) | 0.31 | (0.03) |
| Item 9 | 0.58 | (0.05) | 3.96 | (0.06) | 0.23 | (0.03) |
| Item 10 | 0.48 | (0.05) | 4.04 | (0.05) | 0.31 | (0.03) |
| Risk-taking | | | | | | |
| Item 11 | 0.77 | (0.06) | 3.40 | (0.07) | 0.33 | (0.03) |
| Item 12 | 0.87 | (0.06) | 3.23 | (0.08) | 0.16 | (0.03) |
| Item 13 | 0.77 | (0.06) | 3.29 | (0.07) | 0.33 | (0.03) |
| Autonomy | | | | | | |
| Item 14 | 0.84 | (0.08) | 2.91 | (0.08) | 0.63 | (0.06) |
| Item 15 | 0.99 | (0.07) | 3.49 | (0.09) | 0.19 | (0.06) |
| Item 16 | 0.54 | (0.06) | 3.90 | (0.06) | 0.45 | (0.04) |
| Proactiveness | | | | | | |
| Item 17 | 0.57 | (0.05) | 3.67 | (0.06) | 0.24 | (0.03) |
| Item 18 | 0.66 | (0.06) | 3.54 | (0.07) | 0.30 | (0.03) |
| Item 19 | 0.58 | (0.05) | 3.64 | (0.06) | 0.31 | (0.03) |

Note. Each item and its number correspond to those in Table 2.

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
