# Peer review of "College Students’ Entrepreneurial Mindset: Educational Experiences Override Gender and Major"

_sustainability, doi:10.3390/su12198272_

Round 1

Reviewer 1 Report

I feel that the study is a good one but can be improved more especially from literature review part and improve the conclusion side. 

1. The literature review should be more recent, I would say minimum 70% from the most recent years (latest 3 y).
2. You can make connections with other collectivist countries (GCC for example) - What Determinants Influence Students to Start Their
Own Business? Empirical Evidence from United Arab Emirates Universities (Sustainability) or Innovation and entrepreneurship in Oman and so on. These are relevant to your study.
3. In the conclusion part, needs stronger connection with literature review, own results and the impact of the results (in depth analysis).

Author Response

Thank you for your valuable comments!

Reviewer 2 Report

Summary: The study is aimed (1) to examine measurement invariance of the College Students’ Entrepreneurial Mindset Scale (CS-EMS), (2) to compare the latent and observed means across groups based on gender, major, and educational experiences, and (3) to investigate conditional effects of the three grouping variables.

Abstract: The abstract would benefit by stating the practical implications. Because of this study, what should educators do differently?

Introduction: The introduction would benefit from the inclusion of a research question. In addition, the introduction (or literature review) would benefit by providing a summary of the entrepreneurial mindset assessments currently available (e.g., Entrepreneurial Mindset Profile and Gallup’s BP10 Assessment Tool), with a focus on gaps with current assessment tools.  

Literature Review: Entrepreneurship education and entrepreneurial mindset education is different. These phrases should not be used interchangeably. See Bosman and Fernhaber (2018) text on “Teaching the Entrepreneurial Mindset to Engineers.” In addition, the KEEN (Kern Entrepreneurial Engineering Network) offers a number of resources at www.engineeringunleashed.com. Although the authors highlight the difference between entrepreneurship and the entrepreneurial mindset in section 2.1, they fail to use consistent terminology throughout the manuscript. It is recommended that the term “entrepreneurship” should only be used in section 2.1 for the purpose of differentiating it from the “entrepreneurial mindset.” Thus, section 2.2 should provide a literature review on entrepreneurial mindset education instead of entrepreneurship education; a review of the literature will show these are two very different pedagogical approaches.

Methods: Table 1 should provide a 2x2 showing how many males and females are represented within each major, each grade, and for educational experience. Then, for example, assuming the majority of engineers are male and the majority of non-engineers are female, the authors should speak to this and other generalizations when attempting to conclude males and engineers scored higher. In addition, the methods section should discuss the decision to use CFA instead of EFA.

Results: The results section would benefit by addressing the CFA assumptions.

Discussion: Per the comments related to the methods, the discussion section should speak to the likely high correlation between males and engineers, at a minimum, and its influence on the findings. Then, the title should likely be updated.

Conclusions: Section 4.2 should be relocated to the conclusions.

Author Response

Thank you for your valuable comments!

Reviewer 3 Report

This manuscript tackles an interesting issue in entrepreneurship education, which consists of exploring entrepreneurial mindset. Encouraging this attitude in college students brings benefits not only universities and their students but also to society as a whole. In this regard, I identify two important issues related to the special issue the manuscript was submitted to. On the one hand, it can contribute to the analysis of intrapreneurship within universities, where staff and students work towards entrepreneurship in an entrepreneurial way. On the other hand, equipping students with the necessary knowledge about entrepreneurship, they can contribute to society by solving problems in an entrepreneurial way. Although these strengths are present, I would like to make some suggestions as there is room for improvements. Please find my comments below.

  1. You can improve the discussion of entrepreneurship education tools when saying in the introduction “… That was mainly attributed to the lack of quality assured assessment tools which measure various aspects of educational outcomes in higher education settings. The recently developed and validated assessment – the College Students’ Entrepreneurial Mindset Scale (CS-EMS) [20]–…” In this regard, you can consider other tools such as the Observatory for University Entrepreneurial Activity (OBSEU – Butkouskaya et al., 2020; Urbano et al., 2017) and the Global University Entrepreneurial Spirit Students' Survey (GUESSS –Hahn et al., 2020; Sieger et al., 2016). You can strengthen the importance of your tool by briefly discussing similarities and differences.

Butkouskaya, V., Romagosa, F., & Noguera, M. (2020). Obstacles to Sustainable Entrepreneurship Amongst Tourism Students: A Gender Comparison. Sustainability, 12(5), 1812.

Hahn, D., Minola, T., Bosio, G., & Cassia, L. (2020). The impact of entrepreneurship education on university students’ entrepreneurial skills: a family embeddedness perspective. Small Business Economics, 55, 257–282.

Sieger P., Gruber M., Fauchart E., & Zellweger T. (2016). Measuring the Social Identity of Entrepreneurs: Scale Development and International Validation. Journal of Business Venturing, 31(5), 542-572.

Urbano, D., Aparicio, S., Guerrero, M., Noguera, M., & Torrent-Sellens, J. (2017). Institutional determinants of student employer entrepreneurs at Catalan universities. Technological Forecasting and Social Change, 123, 271-282. Accepted Manuscript.

  1. The introduction section can be improved in two ways. First, you can devote a paragraph to briefly explaining the main findings and contributions of this study to theory/literature, policy, and practice. And second, another paragraph with the paper outline would help readers to know what structure they can find in this paper.

  1. It is perhaps necessary that you state research questions or hypothesis within section 2 (literature review), which might serve readers to identify how you “examine the latent and observed mean differences in the sub-scales of the CS-EMS across studied groups only if at least scalar invariance had been established…,” and “… investigate the conditional effects of the three grouping variables (i.e., gender, major, and experience) using the structural equation modelling framework…”

  1. “Material and Methods” should be in section 3. Subheadings should change as well. Please amend the order of the other headings.

  1. In “Material and Methods,” it would be interesting that you briefly explain why the students’ is 50% more for men than for women. This might be due to the fact that students come from engineering and science.

  1. Within the Appendix, could you please explain what are items 1-19? Are those specific questions of the CS-EMS tool?

  1. By following my third comment, you would also need to bring up those research questions/hypotheses to the “Results” section. Again, this would serve to better guide readers regarding your objectives, literature analysis, and main results.

Author Response

Thank you for your valuable comments!
